# Aggressive surgical approach with major vascular resection for retroperitoneal sarcomas

**Yiyuan Li, Jichun Zhao, Bin Huang, Xiaojiong Du, Hankui Hu, Qiang Guo**⬡*

Division of Vascular Surgery, Department of General Surgery, West China Hospital, Sichuan University, Chengdu , Sichuan Province, China

* rosebud-1@163.com

## Abstract

### Background

En bloc resection of adjacent structures, including major vessels, is often required to achieve negative margins in retroperitoneal sarcoma (RPS). However, the effect of vascular involvement and different reconstruction techniques in patients undergoing vascular resection remains unclear. This study investigated the morbidity, mortality, and long-term survival of patients who underwent an aggressive surgical approach with vascular resection for RPS.

### Methods

We analyzed a prospectively maintained database of patients who underwent surgical resection (with or without vascular resection) for RPS between 2015 and 2020. The primary endpoint was long-term overall survival (OS).

### Findings

The study population comprised 252 patients. Postoperative morbidity, mortality, and OS did not differ significantly between the vascular and no vascular resection groups. Among patients with vascular involvement, those who underwent aggressive surgical approach with vascular resection had a significantly higher OS (66.3 months vs. 25.6 months) compared to those who underwent palliative resection, without an increase in mortality or complication rate. No significant differences were observed in postoperative morbidity, 30-day mortality, or estimated median OS between patients who underwent primary repair and reconstruction.

### Conclusions

In patients with RPS with vascular involvement, an aggressive surgical approach with vascular resection achieved optimal clinical outcomes. Vascular reconstruction techniques had no impact on clinical outcomes.

**Data availability statement:** All relevant data are within the manuscript and its Supporting Information files.

**Funding:** The effort of Dr. Qiang Guo was partially supported by the Natural Science Foundation of Sichuan Province No. 2022NSFSC1388. Other authors do not have any funding source to declare. The funders had no role in study design, data collection and analysis, decision to publish, or preparation of the manuscript.

**Competing interests:** The authors have declared that no competing interests exist.

## Introduction

Aggressive resection is the most effective treatment to reduce local recurrence and improve overall survival (OS) for patients with retroperitoneal sarcomas (RPSs) [1]. Owing to the anatomical characteristics of the retroperitoneal space, RPSs are often asymptomatic and unaccompanied by the compression of adjacent organs [2]. Thus, these tumors are undetected until they are large and closely related to the large blood vessels [2]. RPSs that infiltrate or encase major vessels have been considered unresectable [3]. When the tumor and involved vessels do not have clear tissue margins or cannot be dissected from adjacent major vascular structures, the vessels must be resected and reconstructed for complete surgical resection, posing a more technically challenging surgical approach [4]. However, advances in oncovascular surgery have enabled en bloc resection of the tumors and the involved vessels [5,6].

Some trials have been performed at high-volume institutions to examine the effect of aggressive resection on RPS, but the effectiveness of this approach remains debatable [7–9]. However, few studies have examined the surgical technique and outcomes of patients undergoing resection of major vascular structures for RPS, and most of which are limited to case reports or series with small sample sizes owing to its rarity and technical complexity [10,11]. Limited applications and outcomes that support the feasibility of this technique have been described in these case series. Only two case-control studies have evaluated the effect of an aggressive surgical approach with vascular resection versus tumor resection alone [12,13]. The long-term survival outcomes after vascular resection and tumor resection alone are similar [12,13]. However, one of these studies only enrolled patients with primary retroperitoneal liposarcomas [12], whereas the other only reviewed the results of patients who underwent RPS resection involving only the inferior vena cava [13]. Subgroup analyses were not performed in either study. Moreover, the effect of vascular involvement and different reconstruction techniques in patients undergoing resection of major vascular structures for RPS has not been discussed. The trans-Atlantic Australasian RPS working group recently updated the consensus on managing primary RPS in adults [14]. This consensus recommended that experienced RPS surgeons should evaluate each patient for technical resectability considering technical issues associated with tumor-and patient-related variables. Thus, the estimated prognosis and expected morbidity should be considered before performing oncovascular resection.

The primary aim of this study was to investigate the safety and long-term survival outcomes of an aggressive surgical approach involving vascular resection in patients with RPS. Since there are many different vascular reconstruction techniques (primary repair, ligation, re-implantations, patch repairs, and graft reconstructions), the secondary aim of this study was to compare different reconstruction techniques to determine whether clinical outcomes differed.

## Methods

### Study design and patient population

We queried our prospectively maintained database at West China Hospital, Sichuan University, to identify patients with RPS who underwent surgical resection between 2015 and 2020. Exclusion criteria included patients 1) with visceral sarcomas, benign soft tissue retroperitoneal tumors, or metastatic RPS, 2) who underwent tumor-unrelated or emergency surgery, and 3) with insufficient clinical or histopathological data. The institutional review board of West China Hospital of Sichuan University approved the study protocol (IRB 2022–1674) and informed consent was waived. Patient clinical and pathological data were extracted from the hospital's electronic medical records, anonymized, and stored in a database, maintaining strict confidentiality and being used solely for research purposes on November 24, 2022. As part of

the quality improvement tracking of the analysis of existing data and follow-up survival data, the Clinical Research Unit of West China Hospital, Sichuan University trained and surveyed all the practitioners. The validation of the accuracy and completeness of the database was also supervised by the Clinical Research Unit of West China Hospital, Sichuan University. The study was registered with the Chinese Clinical Trials Registry (ChiCTR) as ChiCTR 2200066276 (https://www.chictr.org.cn/).

The preoperative characteristics, intraoperative findings, and postoperative surgical outcomes were prospectively collected and retrospectively analyzed. Patient demographic and surgical characteristics, including age, sex, body mass index (BMI), comorbid conditions, recurrence status, adjuvant treatment, margin status, organ or vascular resection, transfusion volume, and operation time, were collected. During the study period, for the patients with recurrent RPSs, we only collect the data of the fist admission. Tumor characteristics, such as histological subtype, vascular involvement, and tumor grade, were also collected. The histological subtypes were grouped as follows: well-differentiated liposarcoma (LPS), dedifferentiated LPS, leiomyosarcoma, and others. Relatively rare subtypes, such as malignant peripheral nerve sheath tumors, synovial sarcomas, undifferentiated pleomorphic sarcomas, and solitary fibrous tumors, were grouped together. Tumors were categorized according to the Federation Nationale des Centres de Lutte Contre le Cancer (FNCLCC) grading system as either low-risk (G1), intermediate-risk (G2), or high-risk (G3) [15]. Surgical resections were classified as macroscopically complete (R0/R1) or not (R2). Frailty was quantified using the Eastern Cooperative Oncology Group (ECOG) performance status [16]. Vascular involvement was characterized intraoperatively by the absence of a rim of normal tissue at the tumor-to-vessel interface, tumor encasement, or invasion/infiltration into adjacent major vascular structures. All the surgical procedures were performed by two surgical teams in a single center. The operating surgeon determined the need for vascular/organ resection at the time of surgery. Partial resection or accidental injury to the involved vessels was managed using primary closure or patch repair. Complete resection of the involved vessels was performed using an autologous or polytetrafluoroethylene graft. Primary anastomosis was performed when no tension was observed between the two sides of the involved vessels after complete resection. Ligation of the involved vessels was performed if the collateral circulation was preserved or the supplied organ was concomitantly resected. However, en bloc nephrectomy with ligation of the renal vein/artery was not considered vascular resection. Radiation, chemotherapy, or both were administered as neoadjuvant or adjuvant therapy to a selected group of patients after a multidisciplinary discussion. All patients were followed up every 3 months for the first year and then every 6 months through clinical evaluation and contrast-enhanced computed tomography of the abdomen every 6–12 months after surgery.

## Outcomes

Patients were stratified according to the surgical strategy and categorized as no vascular resection or vascular resection group. Subgroup analyses were conducted on patients with vascular involvement during surgery and those with primary RPS. We also compared different reconstruction techniques and categorized them as primary repair or reconstruction. The primary endpoint was long-term OS. OS was defined as the time from the date of primary tumor diagnosis to the time of death or last contact. Secondary endpoints were recurrence-free survival (RFS) (the time from the date of primary tumor diagnosis to the time of disease recurrence, death, or last contact), postoperative mortality (death within 30 days after surgical resection), postoperative complication (events of Clavien–Dindo classification grade 3 or higher within 30 days after surgical resection) [17], venous thromboembolism (VTE) events (occurrence in patients without VTE before surgery who were newly diagnosed with VTE within 90 days

after the operation), length of hospital stay, and length of intensive care unit (ICU) stay. Patients were screened for deep vein thrombosis (DVT) or pulmonary embolism (PE) if clinically suspected, and all symptomatic VTE events, including DVT and PE, were diagnosed objectively. DVT events were confirmed using venous ultrasonography, and PE was diagnosed using computed tomography.

## Statistical analyses

Continuous patient characteristics were expressed as means with standard deviations or medians (interquartile range [IQR]) and compared using Student's t-test or the Mann–Whitney U test, as appropriate. Categorical variables are expressed as frequency (percentage) and were compared using the chi-square test. Fisher's exact test was used for small, expected frequencies. All the tests were two-tailed. OS and RFS were estimated using the Kaplan–Meier approach and compared between groups using a log-rank test. Statistical significance was set at $p$-value < 0.05. Baseline patient and tumor characteristics with a $p$ value < 0.1 were added to the multivariable regression analysis as possible confounders using the backward stepwise method. Parsimonious models with corresponding risk-adjusted hazard ratios (HRs) and 95% confidence intervals (CIs) were calculated, with prognostic factors identified through backward selection using the likelihood ratio method. All analyses were performed using SPSS Statistics, ver. 28 (IBM Corp., Armonk, NY, USA).

## Results

In total, 292 patients with RPS underwent surgical resection. Of these, 40 patients were excluded for tumor-unrelated or emergency surgery (n = 11), lost to follow-up (n = 13), or incomplete data records (n = 16) (Fig 1). The remaining 252 patients were included in this study. Concomitant organ resections were performed in 40% (102/252) of patients. The median number of resected organs among these patients was two (range, 1–5). Among the 252 patients, 51 were transferred from a local hospital and 21 underwent previous partial resection or biopsy. Furthermore, 44% (n = 110) underwent an aggressive surgical approach with vascular resection, whereas 56% (n = 142) underwent surgical resection without vascular resection. Table 1 summarizes the characteristics of patients who underwent RPS with and without

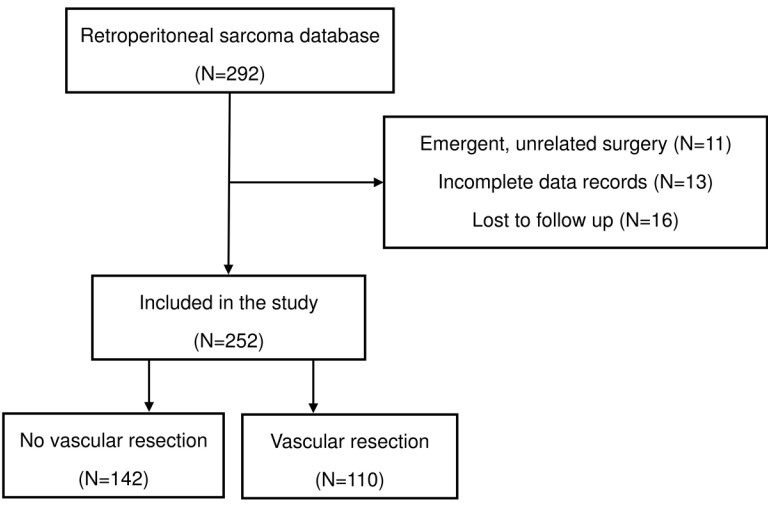

**Fig 1. Flowchart of included patients.**

vascular resection. Patients who underwent vascular resection were more likely to have been transferred from a local hospital (26.4% vs 15.5%; $P = 0.033$), have vascular involvement (79.1% vs 16.9%; $P < 0.001$), a higher FNCLCC grade (59.1% vs 45.1%; $P = 0.027$), history of neoadjuvant radiation (0.9% vs 7.0%; $P = 0.018$), more likely to have undergone organ resection (54.5% vs 29.6%; $P < 0.001$), and a higher transfusion rate (76.4% vs 26.1%; $P < 0.001$).

Table 1 summarizes the postoperative outcomes for each group. No significant differences were observed between the vascular and non-vascular resection group in 30-day mortality (1.4% vs 5.5%) and major complications (4.9% vs 8.2%). However, the no vascular resection group had a significantly lower VTE rate than the vascular resection group (4.2% vs 10.9%; $P = 0.041$). Our analysis demonstrated a 5.3-day shorter mean length of hospitalization (15.9 vs 21.2 days; $P < 0.001$) and a 1.1-day shorter mean length of ICU stay (2.1 vs 3.2 days; $P < 0.001$) in the no vascular resection group compared with the vascular resection group. The median follow-up duration was 29 (IQR, 17–41) months. The 5-year OS rates of patients in the no vascular and vascular resection groups were 64% and 65%, respectively. The estimated median OS did not differ between the vascular and no vascular resection (52.4 months, 95% CI: 45.5–59.4 vs 56.0 months, 95% CI: 50.4–61.7; $P = 0.268$; Fig 2A) groups. Subgroup analyses of different histological subtypes revealed that the estimated median OS did not differ between the vascular and no vascular resection groups in patients with leiomyosarcoma (43.6 months, 95% CI: 35.1–52.0 vs 43.9 months, 95% CI: 27.7–60.1; $P = 0.108$; S1 Fig) or other sarcomas (43.5 months, 95% CI: 31.9–55.1 vs 42.5 months, 95% CI: 32.5–52.5; $P = 0.975$; S2 Fig). However, in patients with well-differentiated LPS, the estimated median OS was significantly shorter in the vascular resection group (49.1 months, 95% CI: 32.9–65.2) than in the no vascular resection group (72.0 months, 95% CI: 64.2–79.7; $P = 0.008$; S3 Fig), and the estimated median OS was significantly longer in the vascular resection group (64.1 months, 95% CI: 55.3–72.8) than in the no vascular resection group (48.3 months, 95% CI: 37.4–59.2; $P = 0.016$; S4 Fig) in patients with dedifferentiated LPS.

S1 Table presents the distribution of different vascular reconstruction techniques and summarizes the proportions of the major vessels involved. In total, 33 patients underwent vascular reconstruction (6 re-implantations, 3 patch repairs, and 24 graft reconstructions), and 77 patients underwent primary repair or ligation. The major vessels involved included 45 inferior vena cava, 28 iliac veins, 22 iliac arteries, 18 abdominal aortae, 23 renal veins, 21 renal arteries, and 38 other vessels.

Table 2 presents the subgroup analyses of patients with vascular involvement. In the entire cohort, 111 patients exhibited vascular involvement during surgery. In this group, 16, 27, and 24 patients had well-differentiated LPS, dedifferentiated LPS, and leiomyosarcoma, respectively. According to the FNCLCC criteria, 6, 35, and 70 cases were classified as grades 1, 2, and 3, respectively. Patients with vascular involvement had a higher proportion of high-grade tumors than those without (63.1% vs 41.8%; $P = 0.001$) and were more likely to have been transferred (27.9% vs 14.2%; $P = 0.007$). Of the 111 patients, 87 underwent an aggressive surgical approach with vascular resection, and 24 underwent palliative resection. The vascular resection approach yielded more organ resection (52.9% vs 8.3%; $P < 0.001$) and transfusion (74.7% vs 8.3%; $P < 0.001$) than the no vascular resection approach, without subsequent differences in FNCLCC grade or histopathological distribution. No differences were observed in the pathological stage distribution. Postoperative morbidity (4.2% vs 8.0%; $P = 0.515$), mortality (4.2% vs 5.7%; $P = 0.762$), VTE rate (0% vs 6.9%; $P = 0.186$), length of stay (18.9 ± 7.9 d vs 21.8 ± 7.6 d; $P = 0.126$), length of ICU stay (2.1 ± 1.7 d vs 3.5 ± 1.7 d; $P = 0.178$) were comparable after aggressive surgical approach and palliative resection. However, the 5-year OS rates of patients in the no vascular and vascular resection groups were 0% and 71%, respectively for those with vascular involvement. The estimated median OS was significantly shorter in the

**Table 1. Baseline demographic and clinical characteristics of the study population.**

| Characteristic | No vascular resection (n = 142) | Vascular resection (n = 110) | P Value |
|---|---|---|---|
| Sex (male) | 67 | 50 | 0.785 |
| Age, median (IQR) | 53 (44–62) | 52 (43–63) | 0.468 |
| BMI, median (IQR) | 23.2 (20.3–26.4) | 23.1 (20.4–25.7) | 0.576 |
| Co-morbidity | | | |
| Hypertension | 15 | 12 | 0.930 |
| Diabetes | 10 | 7 | 0.831 |
| Ischemic heart disease | 3 | 2 | 0.868 |
| Cerebrovascular disease | 1 | 1 | 0.856 |
| ECOG performance status | | | 0.227 |
| 0 | 78 | 52 | |
| 1 | 56 | 46 | |
| 2-3 | 8 | 12 | |
| Transferred | 22 | 29 | 0.033 |
| Recurrent tumor resection | 50 | 39 | 0.968 |
| Tumor size | | | |
| <20 cm | 74 | 50 | 0.294 |
| ≥20 cm | 68 | 60 | |
| Pathology | | | |
| Well-differentiated LPS | 31 | 19 | 0.368 |
| Dedifferentiated LPS | 36 | 32 | 0.507 |
| Leiomyosarcoma | 27 | 21 | 0.988 |
| Others | 48 | 38 | 0.902 |
| FNCLCC grade | | | |
| Low | 18 | 8 | 0.162 |
| Moderate | 60 | 37 | 0.163 |
| High | 64 | 65 | 0.027 |
| Neoadjuvant radiation | 10 | 1 | 0.018 |
| Neoadjuvant chemotherapy | 13 | 6 | 0.270 |
| Postoperative radiotherapy | 15 | 12 | 0.930 |
| Postoperative chemotherapy | 18 | 15 | 0.823 |
| Vascular involvement | 24 | 87 | <0.001 |
| Margin status | | | |
| R0/1 | 114 | 94 | 0.283 |
| R2 | 28 | 16 | |
| Organ resection | 42 | 60 | <0.001 |
| Average no. of organs resected | 1.33 ± 0.87 | 1.57 ± 0.87 | <0.001 |
| Transfusion | 37 | 84 | <0.001 |
| Transfusion volume (ml), median (IQR) | 600 (400–1200) | 1400 (800–2600) | <0.001 |
| Estimated blood loss (ml), median (IQR) | 800 (600–1400) | 1800 (800–3000) | <0.001 |
| 30-d mortality | 2 | 6 | 0.069 |
| Complication | 7 | 9 | 0.294 |
| VTE | 6 | 12 | 0.041 |
| Length of stay (mean, d) | 15.9 ± 8.1 | 21.2 ± 7.8 | <0.001 |
| Length of ICU stay (mean, d) | 2.1 ± 1.5 | 3.2 ± 1.7 | <0.001 |

Abbreviations: IQR, interquartile range; BMI, body mass index; ECOG, Eastern Cooperative Oncology Group; FNCLCC, Federation Nationale des Centres de Lutte Contre le Cancer; LPS, liposarcoma; VTE, venous thromboembolism; ICU, intensive care unit.

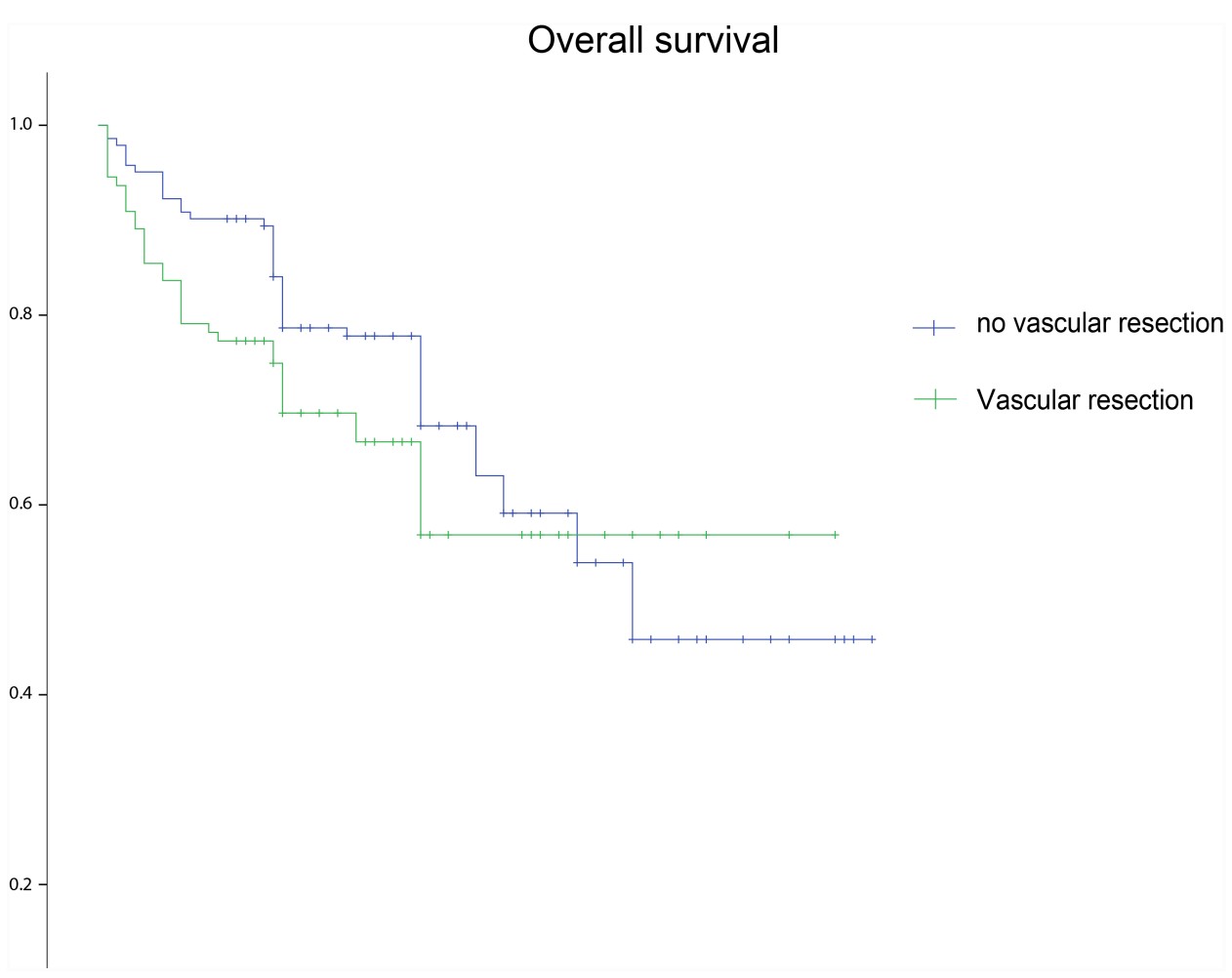

**Fig 2. Overall survival of patients undergoing vascular resection and those without vascular resection for RPS in the (A) entire cohort (B) subgroup analyses cohort (patients with vascular involvement).**

no vascular resection group (25.6 months, 95% CI: 20.0–31.3) than in the vascular resection group (66.3 months, 95% CI: 59.8–72.8; $P < 0.001$; Fig 2B).

Table 3 summarizes the demographic covariates, selected preoperative patient characteristics, and outcomes of patients who underwent surgical resection with vascular resection according to different vascular reconstruction techniques. Compared with the primary repair group, patients who underwent vascular reconstruction were more often women (75.8% vs 45.5%; $P = 0.003$), more likely to have leiomyosarcoma (48.5% vs 6.5%; $P < 0.001$), and had a higher FNCLCC grade (75.8% vs 51.9%; $P = 0.020$). However, no significant difference was observed in postoperative morbidity (6.5% vs 12.1%), 30-day mortality (5.2% vs 6.1%), VTE rate (7.8% vs 18.2%), length of hospital stay (20.9 ± 7.8 d vs 21.8 ± 7.9 d), and length of ICU stay (3.1 ± 1.7 d vs 3.4 ± 1.8 d) between the primary repair and reconstructed groups. Fig 3 depicts the Kaplan–Meier estimates of OS. No significant difference was observed in the estimated median OS between patients following a primary repair (49.5 months, 95% CI: 41.8–57.1) or a reconstruction (52.8 months, 95% CI: 40.5–65.1; $P = 0.754$; Fig 3).

Of the entire cohort, 163 patients underwent primary resection. Subgroup analyses of patients with primary RPSs did not reveal differences in baseline demographics, clinical

**Table 2. Clinicopathological and operative characteristics of patients with vascular involvement.**

| Characteristic | No vascular resection (n = 24) | Vascular resection (n = 87) | P Value |
|---|---|---|---|
| Sex (male) | 13 | 38 | 0.361 |
| Age, median (IQR) | 52(40–61) | 53 (41–63) | 0.532 |
| Recurrent tumor resection | 11 | 33 | 0.484 |
| ECOG performance status | | | 0.692 |
| 0 | 10 | 40 | |
| 1 | 10 | 38 | |
| 2-3 | 4 | 9 | |
| Transferred | 5 | 26 | 0.382 |
| Tumor size | | | |
| <20 cm | 9 | 37 | 0.658 |
| ≥20 cm | 15 | 50 | |
| Histology | | | |
| Well-differentiated LPS | 1 | 15 | 0.106 |
| Dedifferentiated LPS | 5 | 22 | 0.653 |
| Leiomyosarcoma | 5 | 19 | 0.916 |
| Others | 13 | 31 | 0.100 |
| FNCLCC grade | | | |
| Low | 1 | 5 | 0.762 |
| Moderate | 5 | 30 | 0.203 |
| High | 18 | 52 | 0.171 |
| Margin status (R0/1) | | | |
| R0/1 | 0 | 79 | <0.001 |
| R2 | 24 | 8 | |
| Organ resection | 2 | 46 | <0.001 |
| Transfusion | 2 | 65 | <0.001 |
| 30-d mortality | 1 | 5 | 0.762 |
| Complication | 1 | 7 | 0.515 |
| Length of stay (mean, d) | 18.9 ± 7.9 | 21.8 ± 7.6 | 0.126 |
| Length of ICU stay (mean, d) | 2.1 ± 1.7 | 3.5 ± 1.7 | 0.178 |
| VTE | 0 | 6 | 0.186 |

Abbreviations: IQR, interquartile range; ECOG, Eastern Cooperative Oncology Group; LPS, liposarcoma; FNCLCC, Federation Nationale des Centres de Lutte Contre le Cancer; ICU, intensive care unit; VTE, venous thromboembolism.

characteristics, or outcomes (See S2 Table, which presents the clinicopathological and operative characteristics). Predictors and HRs were derived using proportional hazard Cox regression (Table 4). Multivariable analysis revealed that dedifferentiated LPS (HR: 0.231; 95% CI: 0.086–0.622; $P = 0.004$), FNCLCC G3 (HR: 0.279; 95% CI: 0.158–0.493; $P < 0.001$), and R2 resection (HR: 0.373; 95% CI: 0.166–0.838; $P = 0.035$) were associated with worse OS. Dedifferentiated LPS (HR: 0.486; 95% CI: 0.251–0.941; $P = 0.032$), FNCLCC G2 (HR: 0.498; 95% CI: 0.385–0.649; $P = 0.003$) or G3 (HR: 0.325; 95% CI: 0.242–0.435; $P < 0.001$), and R2 resection (HR: 0.452; 95% CI: 0.308–0.719; $P = 0.023$) independently predicted RFS. However, vascular resection did not affect RFS (no vascular resection vs. repair: HR: 0.616; 95% CI: 0.314–1.211; $P = 0.160$; no vascular resection vs. reconstruction: HR: 0.832; 95% CI: 0.419–1.652; $P = 0.599$) or OS (no vascular resection vs. repair: HR: 1.076; 95% CI: 0.461–2.509; $P = 0.866$; no vascular resection vs. reconstruction: HR: 1.021; 95% CI: 0.411–2.536; $P = 0.964$).

**Table 3. Clinicopathological and operative characteristics of patients who underwent vascular resection.**

| Characteristic | Primary repair (n = 77) | Reconstructed (n = 33)* | P Value |
|---|---|---|---|
| Sex (male) | 42 | 8 | 0.003 |
| Age, median (IQR) | 52(43–63) | 53 (40–62) | 0.753 |
| Recurrent tumor resection | 29 | 10 | 0.460 |
| Transferred | 23 | 6 | 0.202 |
| Tumor size | | | |
| <20 cm | 35 | 15 | 0.782 |
| ≥20 cm | 42 | 16 | |
| Histology | | | |
| Well-differentiated LPS | 19 | 0 | 0.002 |
| Dedifferentiated LPS | 29 | 3 | 0.002 |
| Leiomyosarcoma | 5 | 16 | <0.001 |
| Others | 24 | 14 | 0.255 |
| FNCLCC grade | | | |
| Low | 7 | 1 | 0.262 |
| Moderate | 30 | 7 | 0.071 |
| High | 40 | 25 | 0.020 |
| Margin status (R0/1) | | | |
| R0/1 | 64 | 30 | 0.363 |
| R2 | 13 | 3 | |
| Organ resection | 45 | 15 | 0.210 |
| No. of organs resected | 1.62 ± 0.84 | 1.40 ± 0.76 | 0.327 |
| Transfusion | 57 | 27 | 0.215 |
| Transfusion volume (ml), median (IQR) | 1400 (800–2600) | 1300 (1000–3000) | 0.348 |
| 30-d mortality | 4 | 2 | 0.855 |
| Complication | 5 | 4 | 0.324 |
| Length of stay (mean, d) | 20.9 ± 7.8 | 21.8 ± 7.9 | 0.376 |
| Length of ICU stay (mean, d) | 3.1 ± 1.7 | 3.4 ± 1.8 | 0.452 |
| VTE | 6 | 6 | 0.109 |

Abbreviations: IQR, interquartile range; LPS, liposarcoma; FNCLCC, Federation Nationale des Centres de Lutte Contre le Cancer; ICU, intensive care unit; VTE, venous thromboembolism.

*If a patient underwent simultaneous reconstruction and primary repair, the technique was considered as reconstruction.

## Discussion

Our findings suggest that the OS of patients with RPSs undergoing an aggressive surgical approach with or without vascular resection did not differ significantly. In addition, postoperative morbidity and short-term mortality rates were comparable between the two surgical approaches. Our results demonstrate that patients who underwent en bloc RPS resection with major vessels involved had a preponderance of long-term survival compared with those who did not undergo vascular resection, without elevated mortality or complication rates. Furthermore, our multivariable model revealed that histological subtype, pathological grade, and margin status, rather than vascular resection, were independent prognostic factors for OS and DFS. In the vascular resection group, this study demonstrated equivalent postoperative morbidity, mortality, and OS rates between the primary repair and reconstruction groups. The study results contribute to clinical decision-making by showing that vascular resection can be performed with comparable short-and long-term results in patients with RPSs.

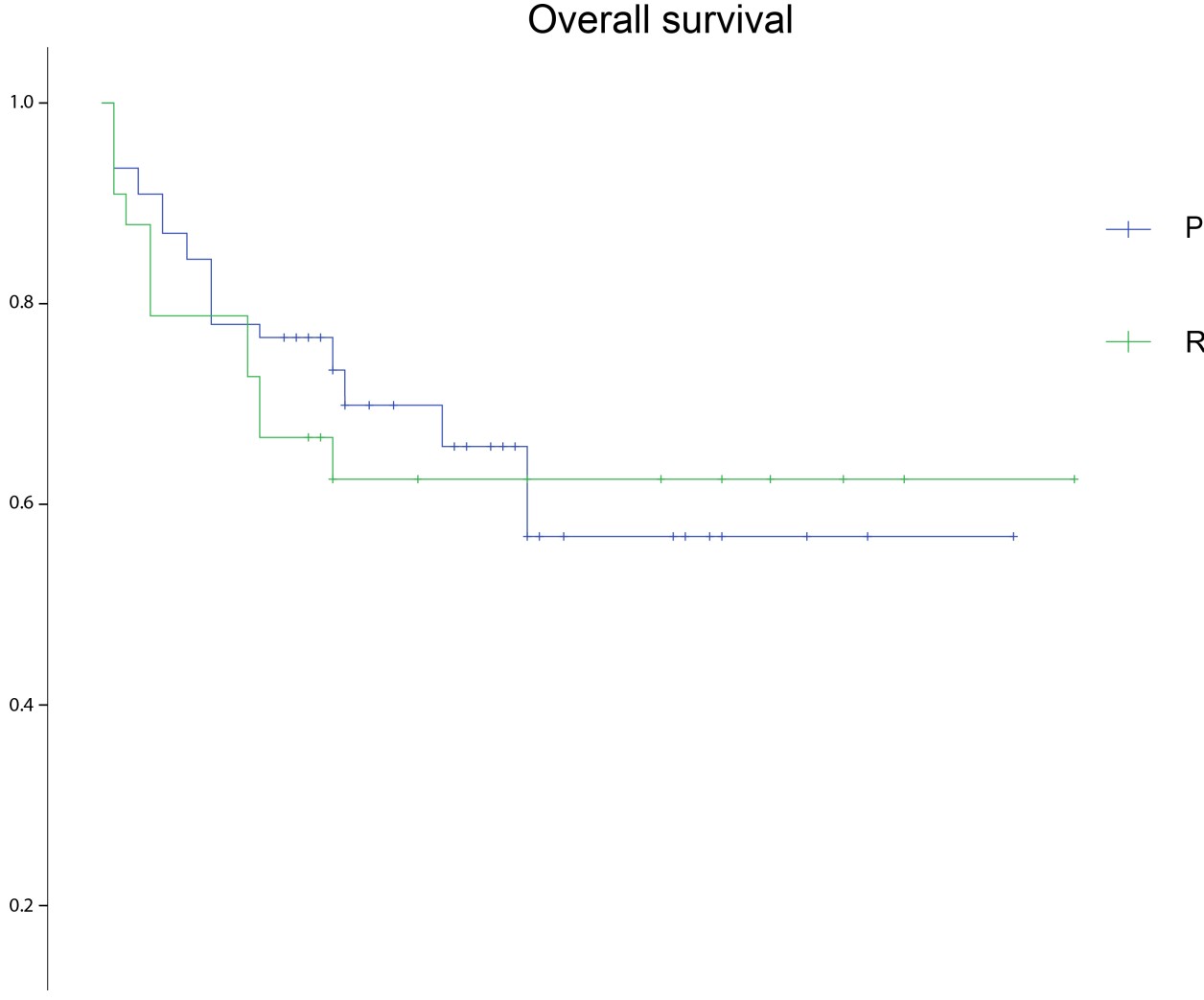

**Fig 3. Kaplan–Meier graph for overall survival of the patients undergoing vascular resection stratified based on different reconstruction techniques (primary repair and reconstruction).**

Our study demonstrated the vascular resection group had a significantly higher blood loss, higher VTE rate, a longer mean length of hospitalization, and a longer mean length of ICU stay than the no vascular resection group, which is supported by previous studies [12,18]. Other studies have also recognized the prognostic value of malignancy grade, surgical margins, and histological type for RPS [19,20]. In this study, patients receiving vascular resection had more radical baseline characteristics because patients who underwent vascular resection were more likely to have been transferred from a local hospital, have vascular involvement, a higher FNCLCC grade, a history of neoadjuvant radiation, and were more likely to undergo organ resection. However, the complete resection rates (R0/R1) after vascular resection were comparable. Thus, although there was a higher blood loss, higher VTE rate, longer length of stay, and length of ICU stay, vascular resection achieved similar OS. The implications of higher blood loss, VTE rates, and extended ICU stays seemed low except for possible higher medical costs.

**Table 4. Multivariate analysis of risk factors for OS and RFS for primary RPSs.**

| Variables | OS | | | RFS | | |
|---|---|---|---|---|---|---|
| | HR | 95% CI | P Value | HR | 95% CI | P Value |
| Age | 1.005 | 0.989–1.021 | 0.570 | 0.994 | 0.981–1.006 | 0.317 |
| Sex | 1.102 | 0.639–1.901 | 0.727 | 0.727 | 0.480–1.101 | 0.132 |
| Pathology | | | | | | |
| Well-differentiated LPS versus Dedifferentiated LPS | 0.231 | 0.086–0.622 | 0.004 | 0.486 | 0.251–0.941 | 0.032 |
| Well-differentiated LPS versus Leiomyosarcoma | 0.650 | 0.314–1.344 | 0.245 | 1.516 | 0.860–2.674 | 0.151 |
| Well-differentiated LPS versus Others | 0.769 | 0.353–1.675 | 0.509 | 0.799 | 0.442–1.444 | 0.457 |
| FNCLCC grade | | | | | | |
| G1 versus G2 | 0.578 | 0.326–1.021 | 0.063 | 0.498 | 0.385–0.649 | 0.003 |
| G1 versus G3 | 0.279 | 0.158–0.493 | <0.001 | 0.325 | 0.242–0.435 | <0.001 |
| Adjuvant therapy | 0.408 | 0.052–3.208 | 0.394 | 0.625 | 0.342–1.134 | 0.235 |
| Vascular resected | | | | | | |
| No vascular resection versus Repair | 1.076 | 0.461–2.509 | 0.866 | 0.616 | 0.314–1.211 | 0.160 |
| No vascular resection versus Reconstructed | 1.021 | 0.411–2.536 | 0.964 | 0.832 | 0.419–1.652 | 0.599 |
| Organ resected | 0.805 | 0.452–1.432 | 0.460 | 1.094 | 0.708–1.690 | 0.687 |
| Margin status | 0.373 | 0.166–0.838 | 0.035 | 0.452 | 0.308–0.719 | 0.023 |
| Transfusion | 1.087 | 0.574–2.059 | 0.797 | 1.488 | 0.884–2.503 | 0.135 |

Abbreviations: OS, overall survival; HR, hazard ratio; CI, confidence interval; RFS, recurrence-free survival; LPS, liposarcoma; FNCLCC, Federation Nationale des Centres de Lutte Contre le Cancer.

Histological examination revealed that patients with vascular involvement had a higher proportion of high-grade tumors than those without vascular involvement, indicating more aggressive behavior. Subgroup analysis of patients with vascular involvement revealed that aggressive surgical approach and palliative resection had similar postoperative morbidity and mortality rates. However, the palliative resection group had a significantly shorter estimated median OS than the vascular resection group. A previous meta-analysis assessing the relative benefits and disadvantages of an aggressive surgical approach with contiguous organ resection in patients with RPS revealed no significant differences in OS and RFS rates between the extended and tumor resection alone groups. However, organ resection did not increase postoperative mortality but increased the complication rate [21]. Further, OS was higher in R0 than in R1 and in R1 than in R2. Therefore, adjacent organs with evidence of direct invasion must be resected to avoid R2 resection. Hence, we suggest that RPS should be indicated for curative-intent resection, even with vascular involvement, and adjacent major vessels should be resected en bloc to avoid R2 resection.

Since recurrent RPS is associated with a worse prognosis than primary RPS [22], and repeat surgery is associated with a significant risk of morbidity and mortality [23], to eliminate the implications of previous procedures, we performed a subgroup analysis of patients with primary RPS. This analysis did not reveal differences in baseline demographics, clinical characteristics, or outcomes. We also performed a multivariable regression analysis of OS and RFS. We found neither primary repair nor reconstruction technique was an independent predictor of OS or RFS. Patients who underwent vascular reconstruction were often women, more likely to have leiomyosarcoma, and had a higher FNCLCC grade than those in the primary repair group. High heterogeneity was observed across previous studies in that most patients who underwent vascular reconstruction had leiomyosarcomas, and most were women [24]. However, no significant difference was observed in the short-term and survival outcomes between the primary repair and reconstruction groups. The possible explanation is that the most

frequently involved major vessel is the inferior vena cava, and aggressive surgical approach and reconstruction of the inferior vena cava can be performed with very low morbidity and mortality and is associated with a low incidence of postoperative symptoms of venous hypertension [25]. Besides, as mentioned above, for retroperitoneal RPS with vascular involvement, regardless of the extent of vascular resection, only histological subtype, pathological grade, and margin status were independent prognostic factors for survival outcomes. Our surgical strategy for retroperitoneal RPS with vascular involvement achieved optimal short-term and survival outcomes, and vascular resection is feasible and safe regardless of the extent of vascular involvement.

The inferior vena cava was the most commonly resected vessel (41%), followed by the iliac vein (25%) and artery (20%), renal vein (21%) and artery (19%), and abdominal aorta (16%). Similar proportions of resected vessels have been reported in other studies [26,27]. However, the proportion of vascular resection and distribution of reconstruction techniques differed from those reported in other studies [26,27]. Based on the results of the previous series, approximately 18–25% of patients with RPS underwent vascular resection, whereas 44% of patients underwent vascular resection in our study [26,27]. In addition to complete resection of involved vessels reconstructed by autologous/synthetic graft or primary anastomose, partial resection or accidental injury of involved major vessels during the exposure of the tumor-vessel border managed with primary closure or ligation was also defined as vascular resection in our study. In addition, we defined vascular involvement as no rim of normal tissue in the tumor-to-vessel interface, tumor encasement, or invasion/infiltration into adjacent major vascular structures, resulting in higher proportions of patients with vascular involvement and patients undergoing vascular resection than that in previous studies. However, the reconstruction rate in this study was 13% (33/252), which is consistent with those reported in previous studies [26,27].

Oncovascular surgery is cancer resection with concurrent ligation or reconstruction of a major vascular structure [28]. Studies have continually supported the feasibility and safety of oncovascular resection [29]. Oncovascular resection enhances resection margins and allows considerable oncologic outcomes in various tumor pathologies with major vascular involvement, including pancreatic cancer, renal cell carcinoma, and cholangiocarcinoma [30]. However, despite the introduction of aggressive surgical approach of RPS with vascular involvement more than two decades ago [27], there have been no advancements in surgical techniques [25], and the incidence of 'unresectability' reported by referral centers remains high, ranging from 10–25% [8]. Upfront unresectability was owing to poor performance status and/or involvement of critical central vascular structures [8]. Here, 20% of the patients were transferred from a local hospital, and 40% of them underwent previous partial resections or biopsies, making our follow-up operations more challenging. Thus, patients with major vessel involvement should be referred to specialized centers and evaluated by an experienced, multidisciplinary team. In addition, guidelines or consensus for the surgical management of RPS are needed to establish new recommendations on aggressive surgical approaches with vascular resection based on the latest evidence, and additional prospective multicenter studies with objective criteria for vascular resection and technical non-resectability are warranted to confirm the reproducibility of the findings of this study.

This study has several limitations. The largest limitation of this study is selection bias due to the retrospective nature of the study. However, while all the surgical procedures were performed by two surgical teams and the surgical policy didn't change over the years, selection bias might be controlled. Furthermore, no propensity score matching was performed, as it would reduce the number of included patients. In this study, vascular involvement was defined as no rim of normal tissue in the tumor-to-vessel interface, tumor encasement, or invasion/

infiltration into adjacent major vascular structures, thus, invasion status and surgical margin status were based on intraoperative findings but was not confirmed via pathology. Besides, the operating surgeon evaluated each patient for technical resectability and determined the need for vascular/organ resection at the time of surgery. It would be subjective for the definition of clinical characteristics of the study population. However, unlike epithelial tumors or adeno-carcinomas, which develop within a single organ, RPSs have poorly defined anatomic borders owing to their large size and multiple central locations. Therefore, assessing clear margin status is impractical. Finally, although we performed a subgroup analysis to investigate the effect of vascular involvement during surgery and in patients with primary RPS, other factors, such as FNCLCC grade, were not evaluated because of the small sample size.

In conclusion, this study reveals that for patients with RPS, aggressive resection surgical approach with vascular resection can be performed safely with an acceptable level of post-operative morbidity and equivalent long-term survival to that without vascular resection. Curative-intent en bloc resection should be performed, and adjacent major vessels with signs of involvement should be resected to avoid R2 resection. Safety and long-term survival results were not associated with the reconstruction techniques. However, the histological subtype, pathological grade, and margin status were independently associated with OS and DFS. Patients with RPS and vascular involvement are best referred to specialized centers and man-aged by an experienced multidisciplinary team.

## Supporting information

**S1 Table. Characteristics of vascular resection.**
(DOCX)

**S2 Table. Clinicopathological and operative characteristics of patients who underwent vascular resection for primary RPSs.**
(DOCX)

**S1 Fig. Overall survival of patients undergoing vascular resection and those without vas-cular resection for leiomyosarcoma.**
(TIF)

**S2 Fig. Overall survival of patients undergoing vascular resection and those without vas-cular resection for other sarcomas.**
(TIF)

**S3 Fig. Overall survival of patients undergoing vascular resection and those without vas-cular resection for well-differentiated LPS.**
(TIF)

**S4 Fig. Overall survival of patients undergoing vascular resection and those without vas-cular resection for dedifferentiated LPS.**
(TIF)

**S1 Data. Data set.**
(RAR)

## Acknowledgments

The authors thank D.Y. Kang, a statistician at the Department of Evidence-based Medicine and Clinical Epidemiology, West China Hospital, Sichuan University, Chengdu, for his assis-tance with the statistical analysis.

## Author contributions

**Conceptualization:** Yiyuan Li, Qiang Guo.

**Data curation:** Yiyuan Li, Jichun Zhao, Bin Huang, Xiaojiong Du, Hankui Hu.

**Formal analysis:** Yiyuan Li, Jichun Zhao, Bin Huang, Xiaojiong Du, Hankui Hu.

**Funding acquisition:** Qiang Guo.

**Investigation:** Yiyuan Li, Jichun Zhao, Qiang Guo.

**Methodology:** Yiyuan Li, Jichun Zhao, Bin Huang, Xiaojiong Du, Hankui Hu, Qiang Guo.

**Project administration:** Yiyuan Li, Jichun Zhao, Bin Huang, Xiaojiong Du, Hankui Hu, Qiang Guo.

**Resources:** Yiyuan Li, Qiang Guo.

**Software:** Yiyuan Li.

**Supervision:** Qiang Guo.

**Validation:** Yiyuan Li.

**Visualization:** Yiyuan Li.

**Writing – original draft:** Yiyuan Li, Qiang Guo.

**Writing – review & editing:** Yiyuan Li, Jichun Zhao, Bin Huang, Xiaojiong Du, Hankui Hu, Qiang Guo.

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
