## [Decision Letter · Decision Letter 0]

20 Dec 2024

PONE-D-24-31199Aggressive surgical approach with major vascular resection for retroperitoneal sarcomasPLOS ONE

Dear Dr. Guo,

Thank you for submitting your manuscript to PLOS ONE. After careful consideration, we feel that it has merit but does not fully meet PLOS ONE’s publication criteria as it currently stands. Therefore, we invite you to submit a revised version of the manuscript that addresses the points raised during the review process.

We look forward to receiving your revised manuscript.

Kind regards,

Paolo Aurello

Academic Editor

PLOS ONE

This research was supported by the Natural Science Foundation of Sichuan Province No. 2022NSFSC1388.  

This research was supported by the Natural Science Foundation of Sichuan Province No. 2022NSFSC1388.

The authors thank D.Y. Kang, a statistician at the Department of Evidence-based Medicine and Clinical

Epidemiology, West China Hospital, Sichuan University, Chengdu, for his assistance with the statistical analysis.

 This research was supported by the Natural Science Foundation of Sichuan Province No. 2022NSFSC1388.

5. Please remove all personal information, ensure that the data shared are in accordance with participant consent, and re-upload a fully anonymized data set. 

Reviewer's Responses to Questions

**Comments to the Author**

1. Is the manuscript technically sound, and do the data support the conclusions?

Reviewer #1: Partly

Reviewer #2: Yes

Reviewer #3: Yes

2. Has the statistical analysis been performed appropriately and rigorously? 

Reviewer #1: I Don't Know

Reviewer #2: Yes

Reviewer #3: Yes

3. Have the authors made all data underlying the findings in their manuscript fully available?

Reviewer #1: Yes

Reviewer #2: Yes

Reviewer #3: Yes

4. Is the manuscript presented in an intelligible fashion and written in standard English?

Reviewer #1: Yes

Reviewer #2: Yes

Reviewer #3: Yes

5. Review Comments to the Author

Reviewer #1: Dear author,

First of all, congratulations on your research. Your sample is consistent. However, I would like to make some considerations.

1. Regarding the method, although the sample is prospective, your work is "retrospective". This aspect should be mentioned.

2. It is also mentioned that this research was registered in the Clinical Trials Registry. Does this mean that there is an ongoing protocol?

3. Shouldn't a stratified analysis have been performed for each histological type?

4. Considering the high number of histological types, which were the other types, should be mentioned and analyzed.

5. The resectability and unresectable criteria depended on the surgeon's assessment. Don't you consider this to be too subjective?

6. The secondary objectives were not very clear.

7. Regarding the R0, R1 resection margin criteria, why was there no statistical difference in the two groups?

8. The high number of R2 patients in the vascular resection group is noteworthy.

This type of disease is very aggressive and does not respond well to adjuvant or neoadjuvant therapy. However, it should be noted that due to the complexity of the surgery, it should only be performed in centers with expertise in oncological surgery and vascular surgery.

Finally, considering the high prevalence of retroperitoneal tumor cases in your center, we also suggest carrying out a prospective and controlled study, analyzing more objective variables.

Thanks.

Reviewer #2: This study evaluates the safety and outcomes of aggressive en bloc resection with vascular involvement in retroperitoneal sarcoma (RPS). It demonstrates that vascular resections improve long-term survival without increasing morbidity or mortality. The findings support curative-intent vascular resections as a feasible strategy for patients with RPS and vascular involvement and are highly relevant for improving survival outcomes and patient care strategies. The manuscript addresses an important and underexplored topic- vascular resections in retroperitoneal sarcoma (RPS), which has significant implications for surgical oncology practice.

This study has used a prospectively maintained database to ensure a robust dataset for analysis. It uses subgroup analyses, including reconstruction techniques, to enhance the depth of the study and allow nuanced conclusions. The manuscript is well-organized, with a logical flow from background to discussion. The data presented in tables and Kaplan-Meier curves is clear. It provides valuable evidence supporting the safety and feasibility of en bloc vascular resections and adds to existing literature by highlighting the potential of aggressive surgical approaches in improving long-term outcomes.

However, I have come across certain issues with the manuscript that need to be addressed to be accepted for publication in this journal.

Some of the major changes are:

1. Although the rationale for the study has been discussed in the introduction section, this section should present a more detailed comparison with recent studies on vascular resections in RPS. Also, the authors should consider highlighting the advancements or gaps addressed by this study.

2. The authors should consider providing more context on how the analysis of baseline differences between vascular and non-vascular resection groups occurs (like in the case of propensity score matching).

3. The authors should elaborately discuss why patients with higher FNCLCC grades were more prevalent in the vascular resection group.

4. Please discuss the implications of higher blood loss, VTE rates, and extended ICU stays for patient outcomes in detail.

5. The authors should highlight the potential strategies to mitigate these risks in clinical practice address the lack of significant differences in overall survival between primary repair and reconstruction techniques and propose hypotheses for these findings.

6. The authors should consider discussing the lack of pathology-confirmed vascular invasion and its impact on findings in a more elaborate manner while discussing the limitations.

7. The authors should mention if any efforts were made to control institutional biases due to the single-center nature of the study.

8. In the figures and tables section, the authors should add a summary table comparing key outcomes across reconstruction techniques for easier reference. Please ensure that all figures (e.g., Kaplan-Meier plots) are annotated with clear legends and relevant p-values for subgroup comparisons.

9. In the discussion section, the authors should include recommendations for multicenter studies or randomized trials to validate findings. They should suggest areas for innovation in surgical techniques or perioperative management to further improve outcomes.

I also found a few minor issues in the manuscript and working on them can improve the manuscript:

1. The authors should reassess sentence structures in the abstract and conclusion to make the findings more concise and impactful.

2. Inconsistent use of terminologies in the manuscript reduces the clarity of the manuscript and might confuse the readers. Therefore, the authors should ensure the use of terminologies consistently throughout the manuscript (for example the authors should either use “en bloc resection” or “aggressive surgical approach” and these terms should not be used interchangeably).

Overall, the manuscript seems promising and has substantial merit in contributing to the field. I believe that addressing these points mentioned above will further strengthen the study’s impact and clarity and make it suitable for acceptance to the journal.

Reviewer #3: 1.Abstract has to be still concise . FNLCC grade need not come in the abstract

2. Vascular involvement in cases has to mentioned not only in tables but also in discussion

3. No difference in OS has to be explained

4.limitations of retrospective data to be highlighted

5.better English

6. PLOS authors have the option to publish the peer review history of their article (what does this mean? ). If published, this will include your full peer review and any attached files.

**Do you want your identity to be public for this peer review?** For information about this choice, including consent withdrawal, please see our Privacy Policy .

Reviewer #1: No

Reviewer #2: No

Reviewer #3: **Yes: ** J.sakthiushadevi

---

## [Author Response · Author response to Decision Letter 1]

4 Feb 2025

Detailed Response to Reviewers

Reply to reviewer #1

Dear reviewer:

We appreciate your attitude to scientific review process and thank you for your comments!

Reviewer #1: Dear author,

First of all, congratulations on your research. Your sample is consistent. However, I would like to make some considerations.

1. Regarding the method, although the sample is prospective, your work is "retrospective". This aspect should be mentioned.

Response:

Yes, our work is "retrospective". We have stated in the limitation part as “this was a retrospective comparative cohort study of prospectively collected data”.

2. It is also mentioned that this research was registered in the Clinical Trials Registry. Does this mean that there is an ongoing protocol?

Response:

The study was registered with the Chinese Clinical Trials Registry (ChiCTR) as ChiCTR 2200066276 (http://www.chictr.org.cn/enindex.aspx). However, it’s not an ongoing protocol, it’s a registered retrospective study.

3. Shouldn't a stratified analysis have been performed for each histological type?

Response:

We have made a stratified analysis for each histological type in the revised paper.

4. Considering the high number of histological types, which were the other types, should be mentioned and analyzed.

Response:

As stated in the second paragraph of method part, in this study, relatively rare subtypes, such as malignant peripheral nerve sheath tumors, synovial sarcomas, undifferentiated pleomorphic sarcomas, and solitary fibrous tumors, were grouped together as ‘others’. Since each of the histological type had less than 20 cases, we didn’t analyzed them seperately.

5. The resectability and unresectable criteria depended on the surgeon's assessment. Don't you consider this to be too subjective?

Response:

Guidelines on the surgical management of RPS are still lacking and remain controversial, owing to its low incidence. For example, the criteria for unresectability remains undefined, and the indication and eligibility for surgical resection vary by medical centre. That is why we conducted this study. We have added a statetment in the limitation part.

6. The secondary objectives were not very clear.

Response:

We have made a revision.

7. Regarding the R0, R1 resection margin criteria, why was there no statistical difference in the two groups?

Response:

In this study, we did not check the tumour margin status routinely. We have added a statetment in the limitation part. Unlike the more common epithelial tumours or adenocarcinomas, which develop within a single organ, RPS can infiltrate multiple surrounding organs owing to their large size and multiple central location. Tumours measuring 20 cm on average have poorly defined anatomic borders, and thus, it would impractical to assess margin status.

8. The high number of R2 patients in the vascular resection group is noteworthy.

Response:

Sixteen of the 110 patients in the vascular resection group were R2 patients. Only 8 of the 16 patients were patients with vascular involvement, the other 8 patients were managed using primary closure or patch repair because of accidental injury to the involved vessels.

This type of disease is very aggressive and does not respond well to adjuvant or neoadjuvant therapy. However, it should be noted that due to the complexity of the surgery, it should only be performed in centers with expertise in oncological surgery and vascular surgery.

Finally, considering the high prevalence of retroperitoneal tumor cases in your center, we also suggest carrying out a prospective and controlled study, analyzing more objective variables. Thanks.

Response:

We are making efforts on carrying out prospective and controlled studies on retroperitoneal tumor cases. Thank you!

Reply to reviewer #2

Dear reviewer:

We appreciate your attitude to scientific review process and thank you for your comments!

Reviewer #2: Reviewer’s Comments

Article: Aggressive surgical approach with major vascular resection for retroperitoneal sarcomas.

This study evaluates the safety and outcomes of aggressive en bloc resection with vascular involvement in retroperitoneal sarcoma (RPS). It demonstrates that vascular resections improve long-term survival without increasing morbidity or mortality. The findings support curative-intent vascular resections as a feasible strategy for patients with RPS and vascular involvement and are highly relevant for improving survival outcomes and patient care strategies. The manuscript addresses an important and underexplored topic- vascular resections in retroperitoneal sarcoma (RPS), which has significant implications for surgical oncology practice.

This study has used a prospectively maintained database to ensure a robust dataset for analysis. It uses subgroup analyses, including reconstruction techniques, to enhance the depth of the study and allow nuanced conclusions. The manuscript is well-organized, with a logical flow from background to discussion. The data presented in tables and Kaplan-Meier curves is clear. It provides valuable evidence supporting the safety and feasibility of en bloc vascular resections and adds to existing literature by highlighting the potential of aggressive surgical approaches in improving long-term outcomes.

However, I have come across certain issues with the manuscript that need to be addressed to be accepted for publication in this journal.

Some of the major changes are:

1. Although the rationale for the study has been discussed in the introduction section, this section should present a more detailed comparison with recent studies on vascular resections in RPS. Also, the authors should consider highlighting the advancements or gaps addressed by this study.

Response:

We have adjusted a part of the content of discussion part to introduction section.

2. The authors should consider providing more context on how the analysis of baseline differences between vascular and non-vascular resection groups occurs (like in the case of propensity score matching).

Response:

No propensity score matching was performed as it would reduce the number of included patients. However, we performed a subgroup analysis to investigate the effect of vascular involvement during surgery and in patients with primary RPS, and we made a stratified analysis for each histological type in the revised paper.

3. The authors should elaborately discuss why patients with higher FNCLCC grades were more prevalent in the vascular resection group.

Response:

We stated in the third paragraph of discussion part as ‘Histological examination revealed that patients with vascular involvement had a higher proportion of high-grade tumors than those without vascular involvement, indicating more aggressive behavior’.

4. Please discuss the implications of higher blood loss, VTE rates, and extended ICU stays for patient outcomes in detail.

We have made a revision in the third paragraph of discussion part.

5. The authors should highlight the potential strategies to mitigate these risks in clinical practice address the lack of significant differences in overall survival between primary repair and reconstruction techniques and propose hypotheses for these findings.

We have made a revision in the fifth paragraph of discussion part.

6. The authors should consider discussing the lack of pathology-confirmed vascular invasion and its impact on findings in a more elaborate manner while discussing the limitations.

Response:

We have made a revision in the limitation part.

7. The authors should mention if any efforts were made to control institutional biases due to the single-center nature of the study.

Response:

As part of the quality improvement tracking of the analysis of existing data and follow-up survival data, the Clinical Research Unit of West China Hospital, Sichuan University trained and surveyed all the practitioners. The validation of the accuracy and completeness of the database was also supervised by the Clinical Research Unit of West China Hospital, Sichuan University. In addition, while all the surgical procedures were performed by two surgical teams and the surgical policy didn't change over the years, selection bias might be controlled. We have made a revision in the method part and the limitation part.

8. In the figures and tables section, the authors should add a summary table comparing key outcomes across reconstruction techniques for easier reference. Please ensure that all figures (e.g., Kaplan-Meier plots) are annotated with clear legends and relevant p-values for subgroup comparisons.

Response:

Results of postoperative morbidity, 30-day mortality, VTE rate, length of hospital stay, and length of ICU stay between the primary repair and reconstructed groups were listed in Table 3. We have made a revision of Fig 2 and Fig 3.

9. In the discussion section, the authors should include recommendations for multicenter studies or randomized trials to validate findings. They should suggest areas for innovation in surgical techniques or perioperative management to further improve outcomes.

Response:

We have made a revision.

I also found a few minor issues in the manuscript and working on them can improve the manuscript:

1. The authors should reassess sentence structures in the abstract and conclusion to make the findings more concise and impactful.

Response:

We have made a revision.

2. Inconsistent use of terminologies in the manuscript reduces the clarity of the manuscript and might confuse the readers. Therefore, the authors should ensure the use of terminologies consistently throughout the manuscript (for example the authors should either use “en bloc resection” or “aggressive surgical approach” and these terms should not be used interchangeably).

Response:

We have made a revision.

Overall, the manuscript seems promising and has substantial merit in contributing to the field. I believe that addressing these points mentioned above will further strengthen the study’s impact and clarity and make it suitable for acceptance to the journal.

Reply to reviewer #3

Dear reviewer:

We appreciate your attitude to scientific review process and thank you for your comments!

Reviewer #3:

1. Abstract has to be still concise. FNLCC grade need not come in the abstract

Response:

We have deleted the content of FNLCC grade in the revised abstract.

2. Vascular involvement in cases has to mentioned not only in tables but also in discussion

Response:

Discussion on vascular involvement was in the third paragraph of the discussion part.

3. No difference in OS has to be explained

Response:

We have made a discussion in the second paragraph of the discussion part.

4. limitations of retrospective data to be highlighted

Response:

We have made a revision in the limitation part.

5. better English

Response:

The manuscript was finally edited by Elsevier Language Editing Services (Order reference No.:ASLEPLUS0451814).

---

## [Editor Report · Decision Letter 1]

13 Feb 2025

Aggressive surgical approach with major vascular resection for retroperitoneal sarcomas

PONE-D-24-31199R1

Dear Dr. Qiang Guo,

We’re pleased to inform you that your manuscript has been judged scientifically suitable for publication and will be formally accepted for publication once it meets all outstanding technical requirements.

Kind regards,

Paolo Aurello

Academic Editor

PLOS ONE

---

## [Editor Report · Acceptance letter]

PONE-D-24-31199R1

PLOS ONE

Dear Dr. Guo,

I'm pleased to inform you that your manuscript has been deemed suitable for publication in PLOS ONE. Congratulations! Your manuscript is now being handed over to our production team.

Kind regards,

on behalf of

Dr. Paolo Aurello

Academic Editor

PLOS ONE